# Pathogenic Carboxyl Ester Lipase (CEL) Variants Interact with the Normal CEL Protein in Pancreatic Cells

**DOI:** 10.3390/cells9010244

**Published:** 2020-01-18

**Authors:** Monica Dalva, Ida K. Lavik, Khadija El Jellas, Anny Gravdal, Aurelia Lugea, Stephen J. Pandol, Pål R. Njølstad, Richard T. Waldron, Karianne Fjeld, Bente B. Johansson, Anders Molven

**Affiliations:** 1Gade Laboratory for Pathology, Department of Clinical Medicine, University of Bergen, N-5020 Bergen, Norway; mda084@gmail.com (M.D.); ida.lavik@gmail.com (I.K.L.); Khadija.Jellas@uib.no (K.E.J.); anny.gravdal@uib.no (A.G.); karianne.fjeld@uib.no (K.F.); anders.molven@uib.no (A.M.); 2Center for Diabetes Research, Department of Clinical Science, University of Bergen, N-5020 Bergen, Norway; pal.njolstad@uib.no; 3Department of Medical Genetics, Haukeland University Hospital, N-5021 Bergen, Norway; 4Department of Pathology, Haukeland University Hospital, N-5021 Bergen, Norway; 5Pancreatic Research Group, Cedars-Sinai Medical Center, Los Angeles, CA 90048, USA; Aurelia.Lugea@cshs.org (A.L.); Stephen.Pandol@cshs.org (S.J.P.); Richard.Waldron@cshs.org (R.T.W.); 6Department of Pediatrics and Adolescent Medicine, Haukeland University Hospital, N-5021 Bergen, Norway

**Keywords:** CEL, BSDL, endocytosis, coexpression, cell viability, pancreatic cell models

## Abstract

Mutations in the gene encoding the digestive enzyme carboxyl ester lipase (CEL) are linked to pancreatic disease. The CEL variant denoted CEL-HYB predisposes to chronic pancreatitis, whereas the CEL-MODY variant causes MODY8, an inherited disorder of endocrine and exocrine pancreatic dysfunction. Both pathogenic variants exhibit altered biochemical and cellular properties compared with the normal CEL protein (CEL-WT, wild type). We here aimed to investigate effects of CEL variants on pancreatic acinar and ductal cell lines. Following extracellular exposure, CEL-HYB, CEL-MODY, and CEL-WT were endocytosed. The two pathogenic CEL proteins significantly reduced cell viability compared with CEL-WT. We also found evidence of CEL uptake in primary human pancreatic acinar cells and in native ductal tissue. Moreover, coexpression of CEL-HYB or CEL-MODY with CEL-WT affected secretion of the latter, as CEL-WT was observed to accumulate intracellularly to a higher degree in the presence of either pathogenic variant. Notably, in coendocytosis experiments, both pathogenic variants displayed a modest effect on cell viability when CEL-WT was present, indicating that the normal protein might diminish toxic effects conferred by CEL-HYB and CEL-MODY. Taken together, our findings provide valuable insight into how the pathogenic CEL variants predispose to pancreatic disease and why these disorders develop slowly over time.

## 1. Introduction

Carboxyl ester lipase (CEL), also known as bile salt-dependent or bile salt-stimulated lipase, is mainly expressed in pancreatic acinar cells and secreted via the ductal tree as a component of pancreatic juice [1]. Once the intrinsic catalytic activity is stimulated by bile salts in the duodenum, CEL participates in the hydrolysis and absorption of cholesteryl esters, dietary lipids, and lipid-soluble vitamins [2,3,4]. It has also been proposed that the enzyme can degrade branched fatty acid esters of hydroxyl fatty acids, a recently discovered group of metabolites with anti-inflammatory and antidiabetic effects [5]. CEL follows the same classical secretory pathway as other pancreatic enzymes [6]. During intracellular transport from the endoplasmic reticulum (ER) to Golgi compartments, CEL is kept in close proximity to intracellular membranes through interactions with the chaperone GRP94 and undergoes several co- and post-translational modifications [7]. In the rough ER, CEL is cotranslationally N-glycosylated on residue Asn210 [8]. The protein is then translocated to the Golgi apparatus where it undergoes extensive O-glycosylation on multiple Ser/Thr residues present in the C-terminal region [6,9]. The latter modification is suggested to prevent CEL from degradation and to protect against self-association caused by exposed hydrophobic regions, thereby increasing the stability and solubility of the enzyme [10,11].

The highly polymorphic *CEL* gene is located on chromosome 9q34 and contains a variable number of tandem repeats (VNTR) region in the last exon [12]. Each repeat consists of nearly identical 33-base pair segments encoding 11 amino acids. The most frequent *CEL* allele in all cohorts studied so far carries 16 repeats, although repeat lengths can vary from 3 to 23 [13,14,15,16,17,18]. We have previously reported that single-base deletions in the *CEL* VNTR cause MODY8 (or CEL-MODY, OMIM 609812), a dominantly inherited syndrome of exocrine and endocrine pancreatic dysfunction [19]. Such deletions lead to a frameshift, changing the C-terminus of CEL into a different, but still repetitive, amino acid sequence. The resulting mutant protein exhibits altered biochemical and cellular properties compared with the normal CEL protein (CEL-WT), and has a higher tendency to aggregate both at the cell surface and intracellularly [20,21]. We have also reported that the pathogenic CEL-MODY protein is reinternalized to a greater extent than CEL-WT and transported to the lysosomes for degradation [22]. Moreover, prolonged exposure to CEL-MODY protein causes reduced cell viability of various cell lines [22].

Several structural variants of the *CEL* locus have been identified, including a pathogenic *CEL* allele designated *CEL-HYB* [23]. In this gene variant, the proximal region of the allele consists of *CEL* sequence, whereas the distal part (including the VNTR) derives from *CELP*, a neighboring pseudogene of *CEL* [12]. The variant is therefore a hybrid allele that encodes CEL-HYB, a CEL-CELP fusion protein. CEL-HYB predisposes to chronic pancreatitis, increasing the risk fivefold. It exhibits reduced lipolytic activity, diminished secretion, accumulation inside the cells, and a tendency to induce autophagy in cellular models [23].

In this report, we examine cellular uptake of CEL-HYB, a process which so far has not been studied. We also extend our previous investigations to pancreatic ductal cells and present evidence of uptake of CEL in human exocrine pancreatic tissue. Finally, we address the observation that both CEL-HYB and CEL-MODY may act dominantly, as affected subjects are heterozygous carriers of these alleles. Until now, however, functional studies have tested the pathogenic CEL variants expressed alone. We therefore also sought to examine interaction effects between CEL-HYB or CEL-MODY and the normal CEL protein.

## 2. Materials and Methods

### 2.1. Plasmids

cDNAs encoding the CEL variants wild-type (WT), hybrid (HYB), and MODY (c.1686delT/p.Val563CysfsX111; previously named MUT) were cloned into the pcDNA3.1/V5-HisB vector (Invitrogen), in-frame with a C-terminal V5/HisB tag. The cloning protocols are described in [21] and [23]. For coexpression experiments, CEL-WT cDNA was inserted in-frame into the p3xFLAG-CMV-13-14 expression vector (Life Technologies, Carlsbad, CA, USA), enabling CEL-WT to be expressed with a C-terminal 3xFLAG epitope.

### 2.2. Antibodies and Reagents

Rabbit polyclonal anti-FLAG (DYKDDDDK; PA1-984B) was from Pierce (Thermo Fisher, Waltham, MA, USA). Mouse monoclonal anti-V5 (R960-25) and F(ab)2-goat anti-mouse IgG-Alexa Fluor 488 (A11017) antibodies were from Invitrogen, Waltham, MA, USA. Mouse monoclonal anti-actin C11 (sc-47778), goat polyclonal anti-GAPDH (sc-20357), mouse monoclonal anti-GAPDH (sc-47724), horseradish peroxidase (HRP)-conjugated donkey anti-mouse IgG (sc-2318), HRP-conjugated mouse IgG kappa binding protein (m-IgGκ BP) (sc-516102), HRP-conjugated donkey anti-rabbit IgG (sc-2305), and HRP-conjugated donkey anti-goat IgG (sc-2020) were all purchased from Santa Cruz Biotechnology, Dallas, TX, USA. Rabbit monoclonal anti-MIST1 (D7N4B) was from Cell Signaling, Leiden, The Netherlands. Mouse monoclonal antibody As20.1, detecting CEL, was generously provided by Prof. O. Hernell (Department of Clinical Sciences, Umeå University, Umeå, Sweden). Rabbit polyclonal anti-CEL (HPA052701) and cycloheximide (CHX) were from Sigma Aldrich, St. Louis, MO, USA. Lipofectamine 2000 transfection reagent, Geneticin (G-418), and staurosporine were all obtained from Invitrogen, RIPA lysis buffer, Triton X-100, and Tween 20 were from Merck Millipore, Burlington, MA, USA. Protease inhibitor cocktail tablets (cOmplete Mini) were from Roche, Basel, Switzerland. Total protein concentration was measured using Pierce BCA protein assay kit (Thermo Fisher). Blots were developed using Amersham ECL Prime Western Blotting Detection Reagent (GE Healthcare, Chicago, IL, USA).

### 2.3. Cell Culture, Transfection, and Drug Treatment

Human embryonic kidney (HEK) 293 cells (from Clontech Laboratories, Mountain View, CA, USA, 632180), mouse 266-6 cells (pancreatic acinar cells; CRL-2151), and human PANC-1 cells (pancreatic carcinoma; CRL-1469), the latter two from American Type Culture Collection, Manassas, VA, USA, were cultured and maintained in Dulbecco’s Modified Eagle’s Medium (DMEM; Sigma-Aldrich) supplemented with 10% fetal bovine serum (FBS; Invitrogen) and 100 units Antibiotics Antimycotic (Sigma-Aldrich). All cell cultures were maintained at 37 °C in 5% CO_2_.

HEK293 cells stably transfected with pcDNA3.1/V5-His plasmids expressing any one of the three CEL variants or a vector with no insert (empty vector; EV) were available from previous studies [21,23]. HEK293 cells stably expressing p3xFLAG-CMV-13-14-CEL-WT were prepared as previously described [21].

For transient transfection, HEK293 cells were cotransfected with pcDNA3.1/V5-His plasmids expressing either CEL-WT, CEL-HYB, or CEL-MODY in a 1:1 ratio with p3xFLAG-CMV-13-14 plasmid expressing CEL-WT-FLAG, using Lipofectamine 2000 transfection reagent (Thermo Fisher) as described by the manufacturer. EV was used as a negative control. To investigate the intracellular stability of CEL proteins, the cotransfected HEK293 cells were treated with 1 µg/mL CHX to inhibit protein synthesis. Treatment started 48 h after transfection, cells being exposed to CHX for 0 to 120 min.

### 2.4. Human Pancreatic Cells and Tissue

Primary human pancreatic acinar cells were obtained from pancreatic cadaveric tissues from organ donors without clinical or histological evidence of pancreatic disease. Briefly, pancreata were digested at City of Hope (Duarte, CA, USA) to separate islets from ductal-acinar trees as previously described [24]. The resulting acinar cells were transferred to Cedars-Sinai Medical Center facilities (Los Angeles, CA, USA) and further processed as recently reported [25]. Acinar cells were cultured for 1, 3, and 16 h with conditioned media from HEK293 cells containing different CEL variants. The study was performed in accordance with regulations and protocols approved by the Institutional Review Boards of the Beckman Research Institute of the City of Hope and the Cedars-Sinai Medical Center (IRB Pro00032114).

Pancreatic specimens available as formalin-fixed, paraffin-embedded (FFPE) tissue blocks were selected from a cohort of quality-controlled pancreatic ductal adenocarcinoma cases studied previously [26]. Patients gave their consent to the study, which was approved by the Regional Ethical Committee of Western Norway (REK Vest 2013/1772) and performed according to the Helsinki Declaration.

### 2.5. Endocytosis

HEK293 cells stably expressing CEL protein variants were grown for 48 h toward 80–90% confluence and the resulting conditioned media were filtered (0.22 µm). Because of varying secretion efficiency (see Appendix A), the conditioned medium was diluted in fresh DMEM to 50% and 33% concentration for CEL-WT and CEL-MODY, respectively, before exposure to recipient cells. Medium containing CEL-HYB was used undiluted. The media were added to primary human acinar cells, nontransfected HEK293, 266-6, or PANC-1 cells for various times. Coendocytosis experiments were conducted similarly, except that the media contained a 1:1 mixture of conditioned medium from CEL-WT-FLAG-expressing cells and from one of the three CEL-V5-tagged protein variants.

### 2.6. SDS-PAGE and Western Blotting

Forty-eight hours after transfection, medium was collected and centrifuged at 13,800× *g* for 5 min at 4 °C. Cells were washed in phosphate-buffered saline buffer (PBS) and lysed in ice-cold RIPA lysis buffer, supplemented with cOmplete Mini protease inhibitor cocktail tablets. The cells were then incubated 30 min on ice and centrifuged at 13800 × *g* for 15 min at 4 °C. The lysate (detergent-soluble) fraction was removed and the pellet washed twice with PBS. To all fractions were added LDS sample buffer and NuPAGE reducing agent (Invitrogen), heated to 56 °C for 15 min (5 min of boiling at 96 °C for the pellet fraction), separated by SDS-PAGE (NuPAGE, 4–12% Bis-Tris gels), and transferred to PVDF membranes using XCell IITM Blot Module system (Invitrogen) according to the manufacturer’s manual. Specific antibodies and enhanced chemiluminescence were used to detect and visualize the proteins of interest. Blots were visualized using a LAS-1000 imager and Image Reader LAS-1000 Pro v2.6 software (Fujifilm, Tokyo, Japan), and quantification of protein bands was performed using Multi Gauge v3.0 software (Fujifilm).

### 2.7. Immunofluorescent Staining and Confocal Microscopy

Cells were grown on poly-L-lysine-coated coverslips, subjected to endocytosis, and fixed for 30 min with 3% paraformaldehyde. Briefly, immunostaining was performed as follows: Wash with PBS, 15 min permeabilization with washing buffer (0.1% (*v*/*v*) Triton X-100 and Tween20 in PBS), wash with PBS followed by 30 min incubation in blocking buffer (washing buffer supplemented with goat serum to a final concentration of 5% (*v*/*v*)), incubation with primary and secondary antibodies (1:200 in blocking buffer) for 1 h and 30 min, respectively, at room temperature or 4 °C overnight, a final wash with washing buffer, followed by rinsing in PBS and water prior to mounting with ProLong Gold Antifade Mountant with DAPI (4’,6-diamidino-2-phenylindole, Molecular Probes, Eugene, OR, USA). Cells were analyzed using Leica SP5 AOBS confocal microscope with 63x/1.4 NA and 40x/1.25 NA HCX Plan-Apochromat oil immersion objectives, ~1.2 airy unit pinhole aperture, and various filter combinations. Images were acquired with 405 diode and argon ion lasers. The obtained images were processed using LAS AF lite (Leica, Wetzlar, Germany), Photoshop CS5 imaging software, and Illustrator CS6 (Adobe, San Jose, CA, USA).

### 2.8. Apoptosis Assay

To analyze apoptosis in the cell cultures, Caspase-3/7 activity levels were measured using a CellEVENT Caspase-3/7 Green detection reagent (Invitrogen) according to the manufacturer’s instructions. The apoptotic assay was performed as described in [22], except that a seeding concentration of 1 × 10^4^ cells/well was used for all cell lines studied. The apoptotic inducer staurosporine (0.5 µM for 4 h) was used as positive control.

### 2.9. Cell Viability

The viability of the cell cultures was analyzed by quantification of ATP generated by metabolic active cells, using a CellTitre Glo luminescent cell viability kit (Promega, Madison, WI, USA) according to the manufacturer’s instructions. The experiment was performed as described in [22], except that 1 × 10^3^ cells/well were seeded for the HEK293 and 266-6 cells, and 1.7 × 10^3^ cells/well for PANC-1 cells.

### 2.10. Immunohistochemistry and In Situ Hybridization

FFPE tissue sections (3–5 µm) were treated as previously described [27] and stained at 4 °C overnight in a 1:100 dilution with polyclonal anti-CEL antibody (Sigma-Aldrich). Using a custom-made CEL-specific probe covering exons 2–7, the RNAscope 2.0HD assay was employed for in situ hybridization according to supplier’s instructions (Advanced Cell Diagnostics, Newark, CA, USA) and [27]. Images were acquired at various magnifications using an MC170HD camera attached to a DM2000LED microscope (Leica) and processed using LAS V4.8 software.

### 2.11. Statistics

To evaluate whether data generated by the pathogenic CEL variants differed significantly from each other or from CEL-WT data, a *t*-test following the Student’s *t*-distribution under null hypothesis was used. The *p*-value was estimated using Microsoft Excel 2010. The significance threshold was set at *p* < 0.05.

## 3. Results

### 3.1. CEL-HYB Endocytosis Affects Apoptosis and Viability of Pancreatic Acinar Cells

We previously reported that both CEL-WT and CEL-MODY can be endocytosed by pancreatic cell lines, and that viability of the exposed cells was negatively influenced by CEL-MODY [22]. To investigate whether also CEL-HYB could be taken up by pancreatic cells and affect their viability, mouse 266-6 acinar cells were treated with conditioned media from CEL-HYB-expressing HEK293 cells and analyzed as in [22]. On average, CEL-HYB was detectable in the cytoplasm of around 11% of the 266-6 cells, a fraction similar to that observed for CEL-WT uptake (6%, Figure 1A,B). In contrast, and as previously reported [22], CEL-MODY was found internalized in 90% of the cells (Figure 1A,B). Hence, CEL-MODY was endocytosed to a significantly higher extent than both CEL-HYB (*p* = 0.004) and CEL-WT (*p* < 0.001). To examine this in a physiologic context, we treated primary human pancreatic acini with conditioned media for various times and analyzed the uptake of different CEL variants by western blotting (Figure 1C). A high cellular uptake was detected for CEL-MODY at 1 h, with a more prominent band being present on the immunoblot at 16 h. Interestingly, CEL-HYB was also internalized at detectable levels at 1 h, although to a lesser extent than the CEL-MODY protein (Figure 1C). For the CEL-WT protein, only a very modest uptake was noted. Intriguingly, the amount of internalized CEL-WT appeared to decrease after 1 h (Figure 1C), suggesting a cellular degradation of the normal protein by the primary acinar cells. These results confirmed and extended our previous findings in cell culture models.

We next analyzed whether long-time exposure to CEL-HYB could affect apoptosis (measured by active caspase-3/7 level) and viability of the 266-6 acinar cells (assayed by ATP level) (Figure 2). Compared with CEL-WT, both CEL-HYB and CEL-MODY induced significantly increased levels of caspase-3/7 (Figure 2A,B, *p* = 0.029 and 0.036), and significantly lower levels of ATP (Figure 2C, *p* = 0.018 and *p* < 0.001). This indicated a possible toxic effect on the pancreatic acinar cell model of the two pathogenic CEL protein variants, where long-time exposure causes increased apoptosis as well as decreased metabolic activity. Although CEL-MODY appeared to have stronger effects than CEL-HYB (Figure 2), the observed differences between these two pathogenic variants were not significant.

### 3.2. CEL Protein Variants are Endocytosed by Pancreatic Ductal Cells

Pancreatic ductal cells secrete bicarbonate and water, thereby aiding the transport and function of digestive enzymes as they pass from the acini into the digestive tract. Internalized CEL protein could potentially disturb this secretion and contribute to tissue damage by provoking leakage of digestive enzymes into the pancreatic parenchyma. Therefore, we investigated whether the CEL protein variants could be endocytosed by a human pancreatic ductal cell model (PANC-1 cells). We tested CEL-WT, CEL-HYB, and CEL-MODY, and found that all variants were taken up by the ductal cells. The variants were detected in the cytoplasm of 9%, 15%, and 33% of the PANC-1 cells, respectively (Figure 3). Thus, the extent of uptake for CEL-WT and CEL-HYB was similar to that observed in acinar cells (and not different from each other), but clearly lower than for CEL-MODY (see Figure 1B). Still, the data showed that the PANC-1 cells had a significantly higher propensity to endocytose CEL-MODY compared with both CEL-WT (*p* = 0.008) and CEL-HYB (*p* = 0.025).

We also evaluated whether CEL might be endocytosed by ductal cells in vivo by analyzing the distribution of CEL mRNA and protein in adjacent sections of morphologically normal pancreatic tissue available from pancreatic cancer patients. As expected, only the acinar cells were positive for CEL mRNA by in situ hybridization (Figure 4A,B). However, immunostaining revealed the presence of CEL protein not only in acinar cells, but also in the epithelium lining the pancreatic duct (Figure 4C,D). In contrast to the acinar staining, the ductal signal was patchy with negative cells interspersed with positive cells of varying staining intensity.

### 3.3. CEL-MODY Can Induce Apoptosis and Reduce the Viability of Pancreatic Ductal Cells

Next, we investigated the effects of the CEL variants on apoptosis and viability in PANC-1 cells after endocytosis (Figure 5). The effects of CEL-WT and CEL-HYB in PANC-1 cells were not significantly different from each other (Figure 5B,C). Exposure to CEL-MODY, however, resulted in increased levels of caspase-3/7 and also a significantly negative effect on viability when compared with CEL-WT (*p* = 0.031 and 0.011, respectively). The effect of CEL-MODY compared with CEL-HYB was significant with regard to levels of active caspase-3/7 (*p* = 0.044), but not viability (Figure 5B,C). The overall pattern indicated that the CEL-MODY protein conferred a stronger toxic effect on the pancreatic ductal cell model than either CEL-HYB or CEL-WT.

### 3.4. The Secretion of CEL-WT is Affected by Pathogenic CEL Variants

In all studies reported so far, the pathogenic CEL variants have been expressed and analyzed alone, that is, without the presence of the normal CEL protein. However, CEL-HYB and CEL-MODY are inherited dominantly as the patients identified are heterozygous carriers of these variants. We therefore aimed to study the impact of CEL-HYB and CEL-MODY when coexpressed with CEL-WT. The CEL variant constructs each included a sequence encoding the C-terminal V5-tag (see Methods). For the coexpression study, CEL-WT cDNA was inserted into the p3xFLAG-CMV-13-14-expression vector backbone. This construct therefore expressed the CEL-WT protein with a FLAG-tag, which will hereafter be referred to as CEL-WT-FLAG. Thus, by using tag-specific antibodies, the V5-tagged CEL protein variants could be distinguished from the FLAG-tagged CEL-WT reference protein in further analyses.

To study the effect of the pathogenic CEL variants on CEL-WT secretion and stability, we transiently transfected HEK293 cells with the CEL-V5 constructs in combination with CEL-WT-FLAG. The expressed proteins in the detergent-soluble cellular fraction (‘lysate’), the insoluble fraction (‘pellet’), and the cell medium were analyzed after 48 h (Figure 6). As previously reported [22,23], an increased amount of CEL-MODY was found in the pellet fraction, whereas CEL-HYB was less secreted into the medium, compared with CEL-WT (Figure 6, left panels). Interestingly, coexpressing CEL-WT-FLAG together with CEL-HYB or CEL-MODY led to increased accumulation of CEL-WT-FLAG in the pellet fraction, as compared with CEL-WT coexpression (Figure 6, right middle panel). There was also a reduction of CEL-WT-FLAG secreted into the medium, whereas the amount of CEL-WT-FLAG found in the lysate fraction seemed to be less affected by the presence of pathogenic CEL variants (Figure 6, right lower and upper panels, respectively).

We then investigated the intracellular stability of CEL-WT-FLAG when coexpressed along with the CEL variants in HEK293 cells. Forty-eight hours after cotransfection, the cells were treated with cycloheximide (CHX) to inhibit protein synthesis and the amounts of the CEL variants and the FLAG-tagged CEL-WT protein were visualized and quantified (Figure 7). Analysis of the soluble fractions showed that intracellular pools of V5-tagged CEL-WT, CEL-HYB, and CEL-MODY decreased during CHX treatment (‘CEL lysate’, Figure 7A–C, left panels).

In addition, intracellular pools of CEL-WT-FLAG protein coexpressed with the V5-tagged CEL variants decreased when protein synthesis was blocked (‘CEL lysate’, Figure 7A–C, right panels; Figure 7D). We observed significantly higher levels of CEL-WT-FLAG in the lysate when coexpressed with CEL-HYB (*p* = 0.03 at time point 0 min) compared to coexpression with the CEL-WT variant (Figure 7D). Notably, the amount of CEL-WT-FLAG in the pellet fractions was significantly higher at the time points 30, 60, and 120 min (*p* = 0.043, 0.034, and 0.006, respectively) when coexpressed with CEL-MODY than when coexpressed with CEL-WT (Figure 7E). This indicated that CEL-MODY affected the secretion of CEL-WT-FLAG and its accumulation within the insoluble fraction. We were unable to detect CEL-WT-FLAG in the medium after CHX treatment (data not shown), though we could detect it in the medium 48 h post-cotransfection (Figure 6). This might be explained by limited FLAG sensitivity of the polyclonal anti-FLAG antibody in combination with low levels of secreted CEL-WT-FLAG due to the relatively short duration of the CHX experiment (0–120 min).

### 3.5. Cointernalized CEL-WT May Diminish the Toxic Effects Conferred by CEL-HYB and CEL-MODY in Pancreatic Cell Line Models

Finally, we performed a coendocytosis experiment in 266-6 cells of either of the two pathogenic CEL protein variants together with CEL-WT. The cells were cultured in a 1:1 mixture of conditioned media from HEK293 cells expressing the different variants, and uptake of CEL protein was confirmed by analyzing the cellular fractions by western blotting (Figure 8A). We observed that all variants were internalized (panel labeled ‘CEL lysate’). Levels appeared similar when the variants were internalized alone or in combinations (compare the ‘EV’ lanes with ‘WT/MODY’ and ‘WT/HYB’). As CEL-HYB is weakly secreted from expressing cells [23], it was present at a very low level in the conditioned medium (note weak band in the panel labeled ‘CEL medium’). Still, it was effectively endocytosed as there was a distinct band appearing in the cell lysate. Notably, only CEL-MODY and CEL-HYB were detected in the insoluble fraction (panel labeled ‘CEL pellet’) in this coendocytosis experiment.

We then assessed cell viability in HEK293, 266-6, and PANC-1 cells after long-time exposure to the pathogenic CEL variants in combination with the CEL-WT protein (Figure 8B). Although exposure to the pathogenic variants trended toward slightly reduced viability, only CEL-HYB reached statistical significance and then only in HEK293 cells (*p* = 0.005). Thus, in three tested cell lines, we observed that coendocytosis appeared less toxic than endocytosis of individual CEL variants (compare with Figure 2C and Figure 5C).

## 4. Discussion

Deposits of aggregated proteins occur in several diseases where proteotoxicity plays a central role. For instance, intracellular aggregates of the islet amyloid polypeptide (IAPP) have been found to clog the ER translocon and to display prion-like properties, ultimately leading to β-cell failure and potentially to type 2 diabetes [28,29]. Relating to neurodegenerative disorders, extracellular aggregates of α-synuclein are likely to be neurotoxic, although receptor-mediated endocytosis followed by lysosomal degradation have been suggested to prevent cells from exposure to the toxic extracellular aggregates [30]. We have previously reported that extracellular CEL-MODY protein could be cleared by endocytosis in pancreatic acinar and β-cell line models, indicating that this pathway might play an important role in MODY8 pathogenesis [22]. Because both disease-associated CEL variants (CEL-MODY and CEL-HYB) display altered biochemical and cellular properties compared with the normal CEL protein, we hypothesized that also CEL-HYB could be internalized by pancreatic cells, subsequently affecting their viability. Moreover, extracellular CEL protein, secreted from the acini, is transported to the duodenal lumen via the ductal tree. This raised the question of whether the bicarbonate-secreting ductal cells might internalize extracellular CEL. We therefore included a pancreatic ductal cell line model in the current study.

Our endocytosis experiments of CEL protein variants in human primary pancreatic acini as well as in pancreatic acinar and ductal cell lines revealed a trend toward slightly higher uptake for CEL-HYB than for CEL-WT. In accordance with previous findings [22], the uptake of CEL-MODY was significantly higher than that observed for CEL-WT (Figure 1 and Figure 3). There were also significantly higher levels of endocytosed CEL-MODY than CEL-HYB protein. Both pancreatic and nonpancreatic cell lines have previously been used to study functional effects of CEL [21,22,31]. Although HEK293 cells are frequently used as an expression system in cell biology and CEL and other proteins are well secreted by this cell line, the pathway in HEK293 is different from the regulated secretion present in pancreatic acinar cells. Moreover, essential post-translational modifications of CEL may be lost. Thus, it was important to investigate CEL in acinar cells such as the murine 266-6 cell line, as these cells provide the transfected and endogenously expressed CEL proteins with a proper exo- and endocytic machinery. The human pancreatic ductal cell line PANC-1 does not express CEL [27]. However, being a cancer cell line, these cells might exhibit properties other than normal ductal cells. In this regard, our analysis of human pancreatic tissue sections is intriguing (Figure 4). The ductal cells were all devoid of *CEL* transcripts. Still, many of them were positive for CEL protein, and we interpret this observation as evidence for ductal uptake of CEL taking place in vivo. The tissue stemmed from patients diagnosed with pancreatic cancer, but *CEL* mRNA is absent also from normal human pancreatic ductal cells [27].

Interestingly, we found a correlation between the degree of endocytosed CEL-HYB and CEL-MODY proteins and cell viability (Figure 1, Figure 2, Figure 3 and Figure 5). A high uptake of aggregated proteins might trigger ER stress and disturb ER homeostasis, ultimately leading to apoptosis [32,33]. This was recently supported by Xiao et al. [31], who reported that intracellular CEL-MODY aggregates generated ER stress and induced apoptosis. The altered physicochemical properties of CEL-MODY were suggested to be responsible for inducing these effects [31]. The major difference between CEL-WT and the two pathogenic variants is the VNTR region [19,23,34]. The different physicochemical properties of CEL-HYB versus normal CEL [23] suggest that internalized CEL-HYB also induces ER stress-associated apoptosis. Moreover, as CEL-HYB is taken up to a smaller extent and reduces cell viability less prominently than CEL-MODY (Figure 1, Figure 2, Figure 3 and Figure 5), we suggest that the pathogenicity of the variants is likely to be linked to their ability to disturb ER homeostasis, rather than the ability to be endocytosed. This renders CEL-MODY with the highest level of pathogenicity, which is in accordance with this variant causing a highly penetrant disease, MODY8 [19]. In our experiments, CEL-HYB generally exhibited cellular effects intermediate between those of CEL-WT and CEL-MODY. Notably, CEL-HYB is only a risk factor for chronic pancreatitis, and most carriers in the population will be unaffected [23]. Hence, the overall results from our functional evaluation appear to reflect the degree of pathogenicity of the two CEL variants.

Evaluating uptake of CEL proteins both in primary acinar cells and in different pancreatic cell lines, acinar cells tended to have a higher propensity to endocytose CEL-MODY than ductal cells (Figure 1 and Figure 3). Despite low uptake, internalization of pathogenic CEL variants by ductal cells is likely of importance for disease progression, as impaired bicarbonate secretion might leave acinar cell secretions as well as aggregated CEL protein stuck in the acini lumen, giving rise to further damage of surrounding cells. Involvement of cell type-specific receptors could be a possible explanation for the observed difference in uptake, as suggested to be the case for aggregated α-synuclein uptake [30]. Multiple endocytic pathways have been revealed [35] and post-translational modifications like glycosylation [36] have been suggested to be important for receptor-mediated endocytosis. Notably, CEL-MODY and, in particular, CEL-HYB have fewer predicted sites for O-glycosylation than the WT protein. This is reflected in a less complex pattern of O-glycans attached to the pathogenic proteins [37], which again could be of relevance for endocytosis and pathogenicity. Moreover, modifications of the CEL VNTR region by ABO blood group antigens [27] might affect the distinct affinity and uptake kinetics of each cell type toward CEL protein variants, by altering the receptor-binding properties of CEL.

All functional CEL studies have until now analyzed CEL-HYB and CEL-MODY expressed alone, despite the fact that the affected subjects are heterozygous carriers. We therefore aimed to investigate whether CEL-HYB or CEL-MODY could influence the intracellular properties of CEL-WT, using a coexpression model. Our cotransfection results concur with previous findings [21,22,23], as both CEL-HYB and CEL-MODY displayed impaired secretion and accumulated intracellularly, indicating that the coexpression model is valid. Using this model, we found that CEL-HYB and CEL-MODY affected the intracellular fate of a FLAG-tagged CEL-WT protein (Figure 6 and Figure 7), as the latter was found to be more intracellularly accumulated when coexpressed with a pathogenic CEL variant than with CEL-WT-V5. Intriguingly, the rate of CEL secretion has been suggested to positively correlate with the cells’ capability to O-glycosylate the CEL VNTR region [9], partially explaining the impaired secretion of CEL-HYB and CEL-MODY, as the VNTR regions of these enzymes are shorter and potentially less O-glycosylated compared to CEL-WT [19,23,27]. However, it does not explain how the pathogenic CEL variants affect the intracellular fate of CEL-WT. As previously suggested, intracellular CEL-HYB and CEL-MODY aggregates may disrupt ER homeostasis, thereby impairing protein secretion from affected cells. This might cause retention or intracellular accumulation of CEL-WT, as we observed in this study. Additionally, we have previously suggested that the altered physicochemical properties of the CEL VNTR region could disturb short- and long-range protein interactions [21], allowing the pathogenic CEL variants to interact with CEL-WT and subsequently affect its intracellular fate.

A limitation of our study is that the experimental setting, even if various cell types are investigated, cannot mirror the complex interactions that take place in vivo. Analyzing endocytosis of CEL variants in pancreatic organoids and in isolated islets would therefore represent a natural next step. Nevertheless, we believe that our findings may be important for understanding the underlying disease process induced by the pathogenic proteins CEL-HYB and CEL-MODY. We showed that not only CEL-MODY but also CEL-HYB can be endocytosed by human primary acinar cells as well as pancreatic cell lines, negatively affecting viability of the exposed cells. In addition, by using a coexpression model, we found that CEL-HYB and CEL-MODY affected the intracellular fate of CEL-WT, and that cellular toxicity of the pathogenic variants is reduced by the presence of the normal CEL protein. In this regard, it would be interesting to evaluate in follow-up experiments whether the presence of the WT protein also decreases the ER stress response induced by CEL-MODY [31].

The finding of a possible protective effect of CEL-WT is pertinent to the observation that pancreatic disease associated with both CEL-HYB and CEL-MODY occurs in the heterozygous state and develops slowly over time [19,23,38,39]. We suggest a common disease mechanism for these variants, possibly involving the endocytic pathway, in which proteotoxicity due to intracellular protein aggregation and accumulation is pivotal for disease development. Since CEL-HYB is genetically less penetrant than CEL-MODY, we further propose that CEL-HYB causes pancreatic disease in conjunction with other risk factors. To define these additional factors that, together with the underlying susceptibility conferred by CEL-HYB, precipitate chronic pancreatitis should be addressed in further studies.

## Figures and Tables

**Figure 1 cells-09-00244-f001:**
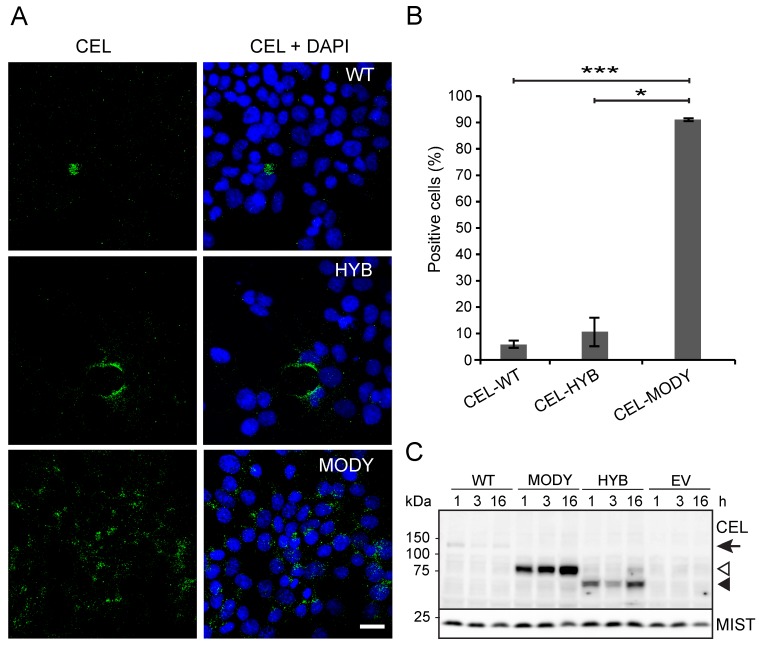
Endocytosis of secreted CEL-HYB protein in pancreatic acinar cells. (**A**): 266-6 cells were cultured for 30 min in conditioned medium from HEK293 cells stably expressing CEL-WT, CEL-HYB, or CEL-MODY, followed by fixation and immunostaining with the CEL-specific antibody As20.1. Scale bar is 20 µm. (**B**): Quantification of the experiment showed that CEL-WT and CEL-HYB were on average internalized by 6% and 11% of the 266-6 cells, respectively, whereas CEL-MODY was taken up by 90% of the cells. For each variant, a total of 700–740 cells were counted in three independent experiments. Error bars represent SEM (n = 3); * and *** denote *p* < 0.05 and *p* < 0.001, respectively. (**C**): Human primary pancreatic acini were cultured in conditioned medium from stably transfected HEK293 cells (expressing CEL-WT, CEL-HYB, or CEL-MODY) for 1, 3, and 16 h. Immunoblots from lysates show protein levels of internalized CEL protein variants at the different time points. Expression of the transcription factor MIST1 was used as loading control. Arrow indicates the position of CEL-WT, open arrowhead CEL-MODY, and filled arrowhead CEL-HYB. One representative image of three experiments is shown.

**Figure 2 cells-09-00244-f002:**
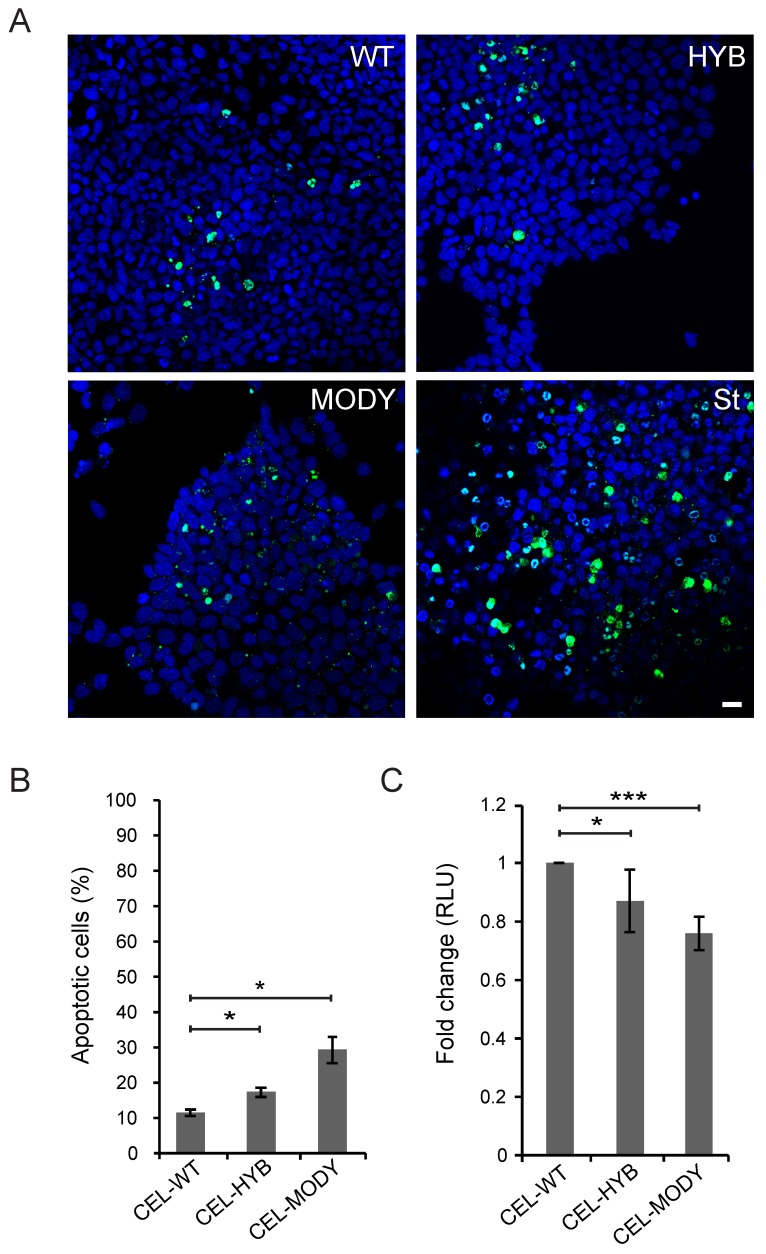
Effect of endocytosed CEL-HYB on apoptosis and viability of pancreatic acinar cells. 266-6 cells were cultured for seven days in the presence of conditioned medium from HEK293 cells stably expressing CEL-WT, CEL-HYB, or CEL-MODY. (**A**): Apoptosis was assessed by visualization of caspase-3/7 activity in the 266-6 cells exposed to different CEL protein variants. As positive control, cells were treated with 0.5 µM staurosporine (St). Scale bar is 50 µm. (**B**): Quantification of the experiment showed that CEL-HYB caused elevated induction of apoptosis compared with CEL-WT, but lower than for CEL-MODY. For each variant, a total of 500–700 cells were counted in three independent experiments. (**C**): Cell viability was analyzed by measuring the intracellular level of ATP. 266-6 cells exposed to CEL-HYB exhibited reduced viability compared to cells exposed to CEL-WT, but not as much as for CEL-MODY. The experiment was repeated three times, each with 9–12 parallels per condition. Error bars represent SEM (n = 3); * and *** denote *p* < 0.05 and *p* < 0.001, respectively. RLU, relative light units.

**Figure 3 cells-09-00244-f003:**
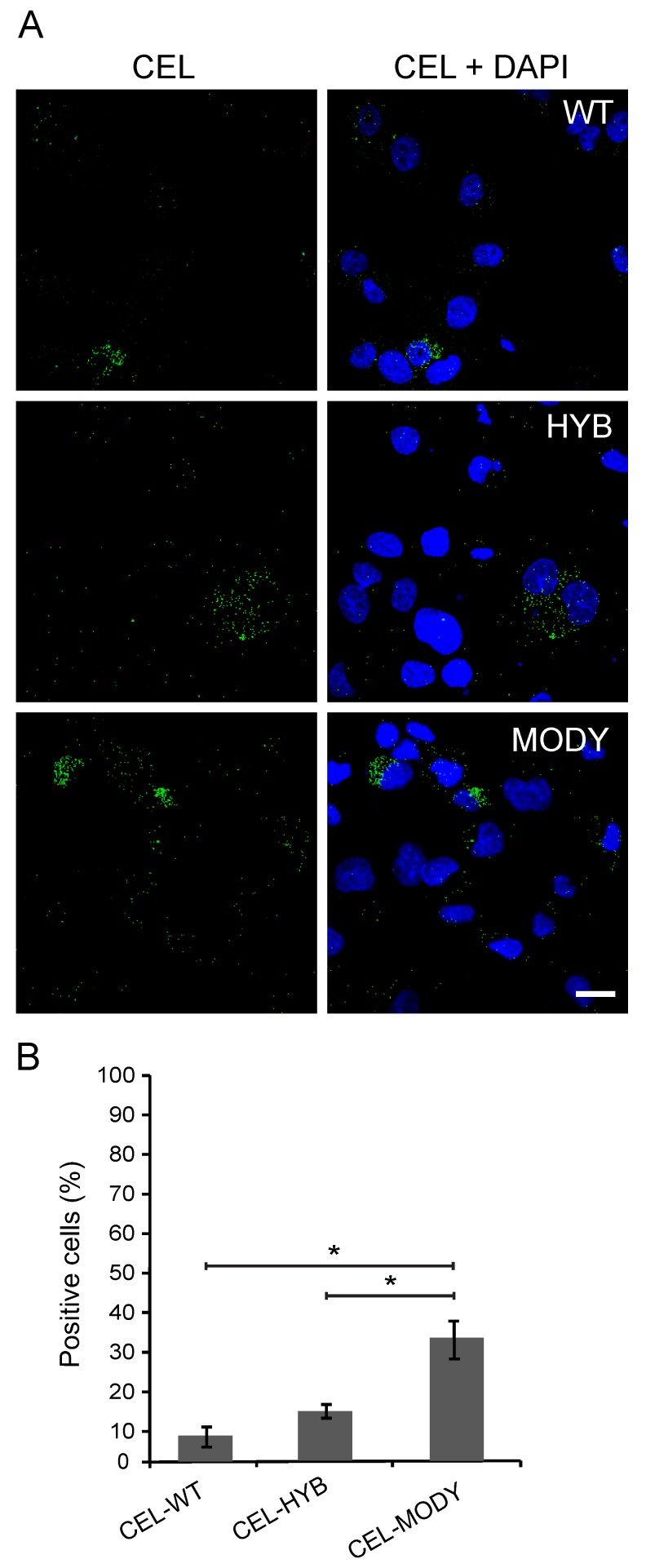
Endocytosis of secreted CEL protein variants in pancreatic ductal cells. (**A**): PANC-1 cells were cultured for 3 h in conditioned medium from HEK293 cells stably expressing CEL-WT, CEL-HYB, or CEL-MODY, followed by fixation and immunostaining using the CEL-specific antibody As20.1. Scale bar is 20 µm. (**B**): Quantification of the experiment showed that all CEL variants could be detected in the cytoplasm of the PANC-1 cells with CEL-MODY having significantly higher frequency of positive cells than CEL-WT and CEL-HYB. For each variant, a total of 450–500 cells were counted in four independent experiments. Error bars represent SEM (n = 4); * denotes *p* < 0.05.

**Figure 4 cells-09-00244-f004:**
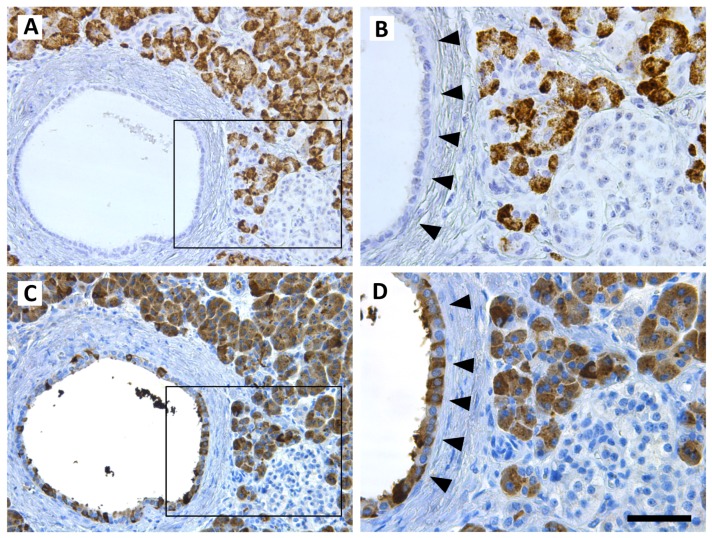
Evidence for in vivo reuptake of secreted CEL protein in human pancreatic ductal cells. Low (**A**) and high (**B**) magnification of in situ hybridization staining in morphologically preserved pancreatic tissue from a case of pancreatic ductal adenocarcinoma. The acinar cells were strongly positive for CEL mRNA transcripts, whereas ductal (arrowheads) and islet cells (lower right corner) were negative. Low (**C**) and high (**D**) magnification of an adjacent tissue section that was immunostained for CEL protein expression with an anti-CEL antibody (Sigma-Aldrich). Acinar cells were positive and islet cells negative as in (**A**,**B**). Positive, patchy staining of varying intensity in ductal cells (arrowheads) indicated that CEL protein might have been endocytosed from the pancreatic juice by these cells. Scale bar is 100 µm for low magnification (**A**,**C**) and 50 µm for high magnification (**B**,**D**).

**Figure 5 cells-09-00244-f005:**
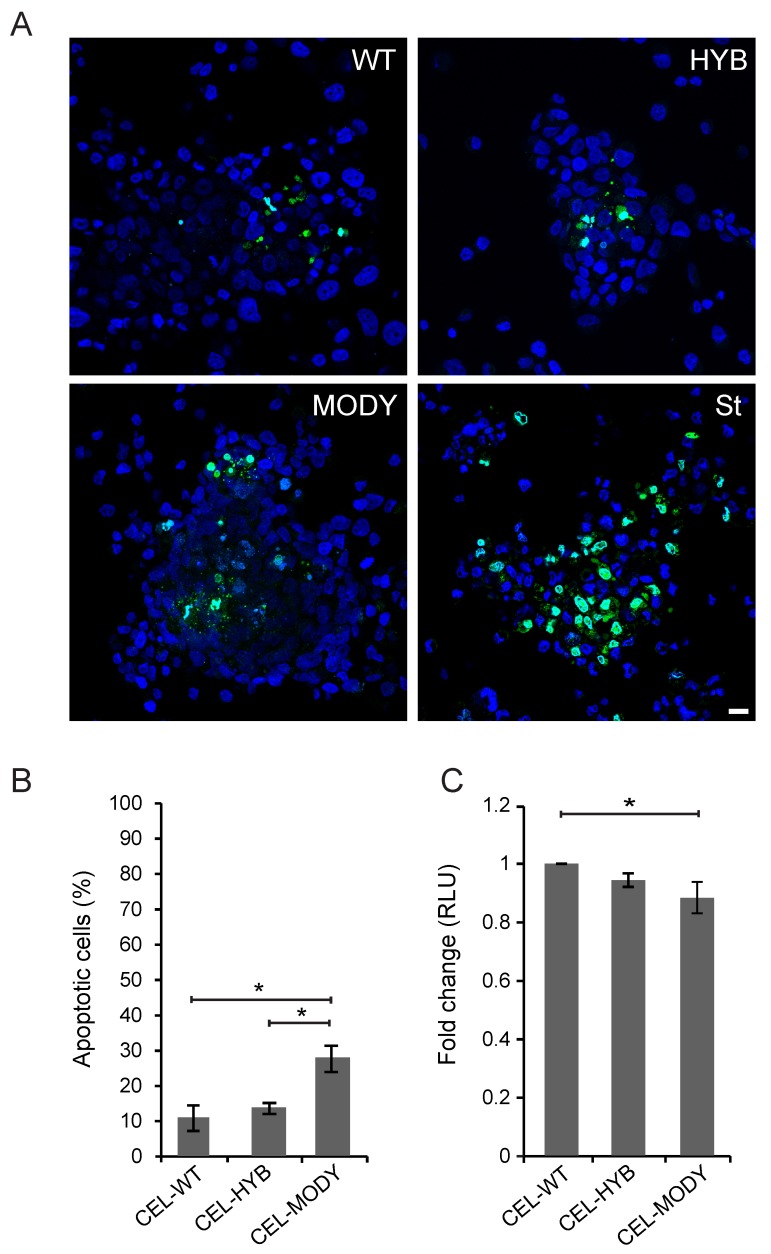
Effect of endocytosed CEL variants on apoptosis and viability of pancreatic ductal cells. PANC-1 cells were grown for seven days in the presence of conditioned medium containing CEL-WT, CEL-HYB, or CEL-MODY. (**A**): Apoptosis was assessed by visualization of caspase-3/7 activity in the PANC-1 cells exposed to different CEL protein variants. As positive control, cells were treated with 0.5 µM staurosporine (St). Scale bar is 50 µm. (**B**): Quantification of the experiment showed that CEL-MODY displayed significantly elevated caspase-3/7 activity compared with CEL-WT and CEL-HYB. For each variant, a total of 300–400 cells were counted in three independent experiments. Error bars represent SEM (n = 3); * denotes *p* < 0.05. (**C**): Cell viability was analyzed by measuring the intracellular level of ATP. PANC-1 cells exposed to CEL-HYB exhibited reduced but nonsignificant viability compared with cells exposed to CEL-WT, whereas CEL-MODY resulted in significantly reduced viability. The experiment was repeated two times, each with 11–12 parallels per condition. Error bars represent SEM (n = 2); * means *p* < 0.05. RLU, relative light units.

**Figure 6 cells-09-00244-f006:**
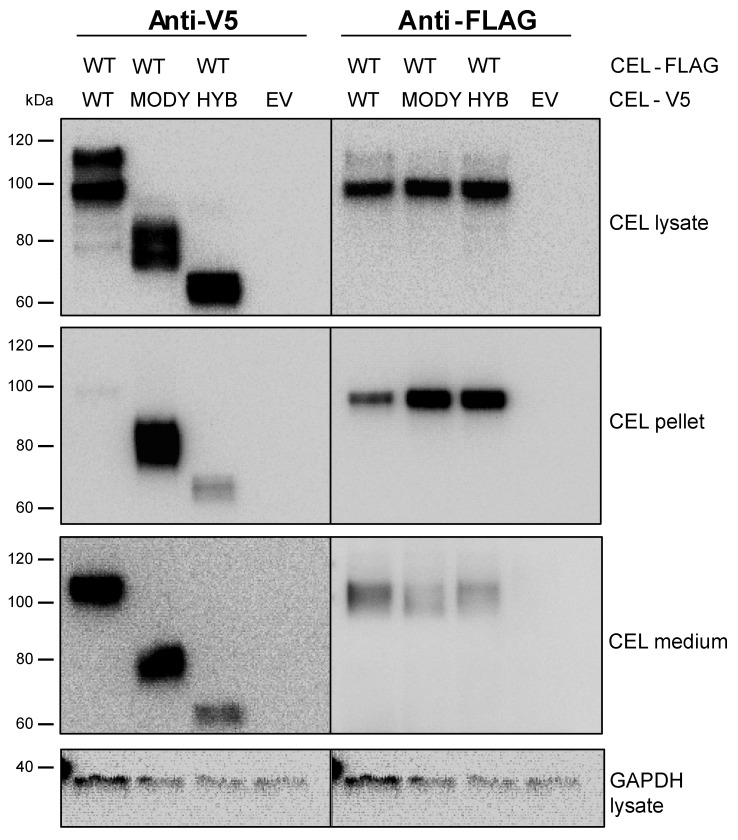
Coexpression and secretion of CEL protein variants in transiently transfected HEK293 cells. The cells were cotransfected with a FLAG-tagged CEL-WT plasmid and plasmids encoding V5-tagged CEL-WT, CEL-HYB, or CEL-MODY. Empty vector (EV) was used as negative control. Soluble (‘lysate’), insoluble (‘pellet’), and medium fractions were collected 48 h after cotransfection and analyzed by western blotting using anti-FLAG and anti-V5 tag-specific antibodies. GAPDH in the lysate fraction was used as loading control. The image shown is representative of three independent experiments.

**Figure 7 cells-09-00244-f007:**
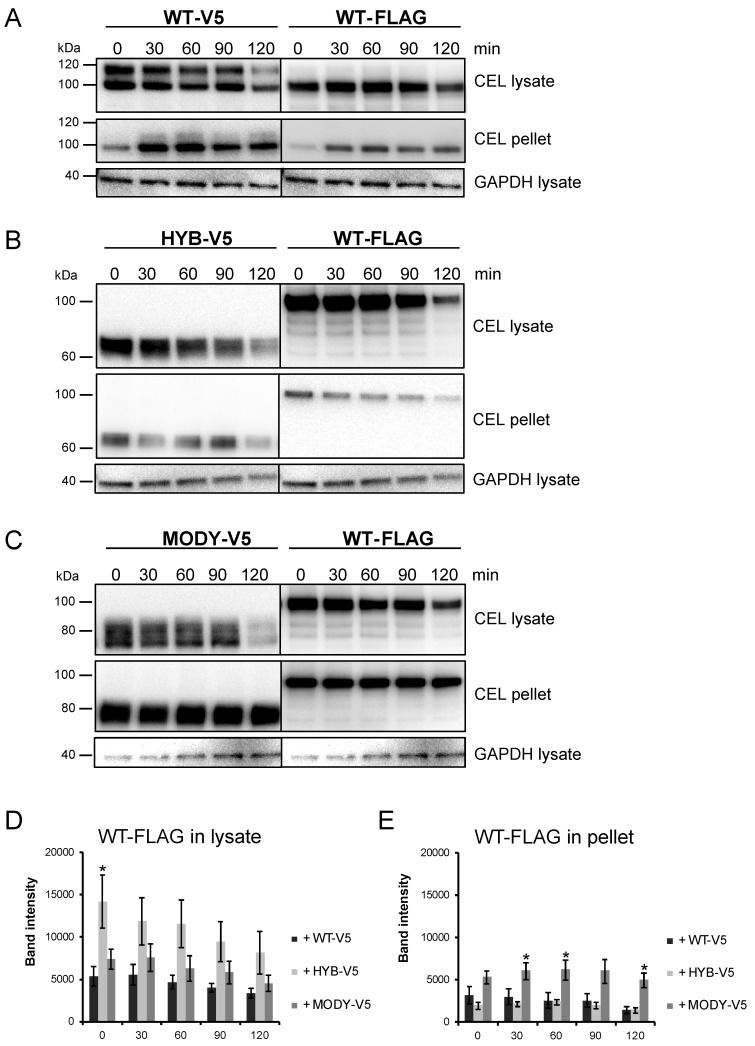
Cellular clearance of normal CEL protein in the presence of pathogenic CEL variants. (**A**–**C**): HEK293 cells were cotransfected with a FLAG-tagged CEL-WT plasmid and plasmids encoding V5-tagged CEL-WT, CEL-HYB, or CEL-MODY. After 48 h, the cells were treated with 1 µg/mL cycloheximide for 0, 30, 60, 90, and 120 min. The lysate and pellet fractions were isolated and analyzed by anti-FLAG and anti-V5 tag-specific antibodies. GAPDH in the lysate fraction was used as loading control. (**D**,**E**): Bands in A–C (WT-FLAG panels) were quantified, normalized against GAPDH, and plotted as values for band intensity (arbitrary units). The experiment was repeated three times, each time with 2–3 parallels per set-up. Data are shown as mean values, the error bars representing SEM (n = 3). * denotes *p* < 0.05 when value is compared to the value for CEL-WT-FLAG coexpressed with CEL-WT.

**Figure 8 cells-09-00244-f008:**
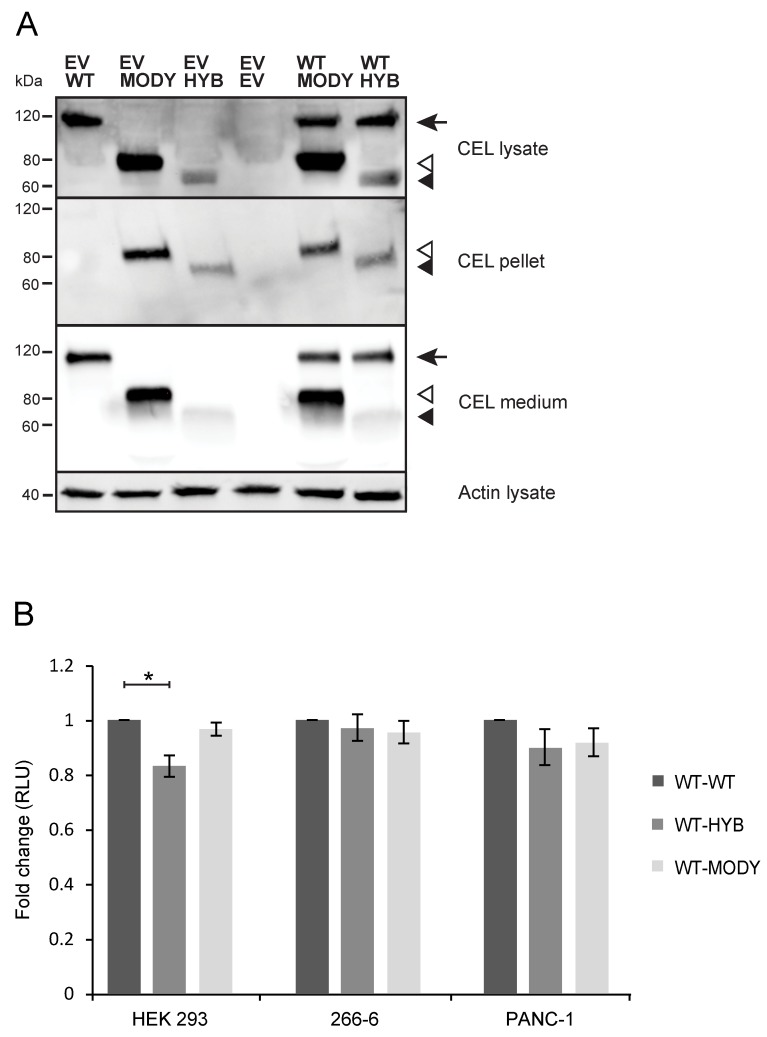
The effect of coendocytosis of secreted CEL protein variants on cell viability. (**A**): Untransfected 266-6 cells were cultured for 24 h in a 1:1 mixture of conditioned medium from two different HEK293 cell lines stably expressing either CEL-WT, CEL-MODY, CEL-HYB, or EV in the combination indicated on top of the figure. The medium, lysate, and pellet fractions were analyzed by western blotting using an anti-V5 tag-specific antibody. Actin in the lysate fraction was used as loading control. The arrow indicates the position of CEL-WT, open arrowhead shows CEL-MODY, and filled arrowhead CEL-HYB. The experiment was repeated three times with similar results, and one representative image is shown. (**B**): HEK293, 266-6, and PANC-1 cells were grown for seven days in the presence of conditioned medium containing CEL-WT-FLAG and either CEL-WT, CEL-HYB, or CEL-MODY. Cell viability was measured as intracellular ATP content. The experiment was repeated four times for each cell line, with 7–12 parallels per condition. Data are shown as mean values, the error bars representing SEM (n = 4). * denotes *p* < 0.05 when compared with CEL WT-FLAG coendocytosed with CEL-WT-V5. RLU, relative light units.

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
