# Peer review of "Pathogenic Carboxyl Ester Lipase (CEL) Variants Interact with the Normal CEL Protein in Pancreatic Cells"

_cells, 2020, doi:10.3390/cells9010244_

Round 1

Reviewer 1 Report

Dalva et al. investigated the effect of pathogenic CEL variants (CEL-MODY, CEL-HYB) on pancreatic acinar and ductal cells. The experiments are well designed and the conclusions are supported by the results.

Please find below my comments, suggestions, questions.

Assuming, that the conditioned media from HEK293 cells transfected with different CEL constructs were diluted due to the secretory differencies of these variants, it would be nice to see as a Supplementary file the actual secretion of these constructs by HEK293 cells or please describe in Methods section why were these media diluted. Please correct in the Figure 1 description the amount of internalized CEL-HYB (from 15% to 11% - in text on Page 5 is 11%). In case of co-endocytosis experiments, has apoptosis been investigated? Do the results correlate with the cell viability data? Is there any experimental evidence that the endocytosed CEL variants generate ER stress? It would be nice to prove the activation of the ER machinery in both cases.

Author Response

Thank you for reviewing. Please see Attachment.

Kind regards

Reviewer 2 Report

The present MS by Monica Falva et al. focuses on pathophysiological mechanisms of two variants of CEL. The study is a comprehensive analyses that uses different experimental methods to approach this aim. The results bring new insights and are interesting for pancreatologist. Two major points must be adressed: the in vitro models seem to be limited (Panc-1 cells as model for ductal cells; mice acinar cells) and the experimental setting can not mirror the complex interactions that take place in vivo. I think the quality of the study design outweighs these criticisms, however I would mention it in the discussion section.

Minor points:

Introduction:

-Some of the references are quite old, aren´t there more recent references?

- "CEL is a highly polymorphic gene" this sentence is cited twice.

Author Response

(The authors gave the same response as above.)
